# Local Anesthetics and Recurrence after Cancer Surgery-What’s New? A Narrative Review

**DOI:** 10.3390/jcm10040719

**Published:** 2021-02-11

**Authors:** Sarah D. Müller, Jonathan S. H. Ziegler, Tobias Piegeler

**Affiliations:** Department of Anesthesiology and Intensive Care, University Hospital Leipzig, Liebigstrasse 20, 04103 Leipzig, Germany; sarah.mueller2@medizin.uni-leipzig.de (S.D.M.); jshz@posteo.de (J.S.H.Z.)

**Keywords:** local anesthetics, cancer, recurrence, inflammation, metastasis

## Abstract

The perioperative use of regional anesthesia and local anesthetics is part of almost every anesthesiologist’s daily clinical practice. Retrospective analyses and results from experimental studies pointed towards a potential beneficial effect of the local anesthetics regarding outcome—i.e., overall and/or recurrence-free survival—in patients undergoing cancer surgery. The perioperative period, where the anesthesiologist is responsible for the patients, might be crucial for the further course of the disease, as circulating tumor cells (shed from the primary tumor into the patient’s bloodstream) might form new micro-metastases independent of complete tumor removal. Due to their strong anti-inflammatory properties, local anesthetics might have a certain impact on these circulating tumor cells, either via direct or indirect measures, for example via blunting the inflammatory stress response as induced by the surgical stimulus. This narrative review highlights the foundation of these principles, features recent experimental and clinical data and provides an outlook regarding current and potential future research activities.

## 1. Introduction

Local anesthetics (LA) are well-known substances and a mainstay of anesthesia since the introduction of cocaine in 1884. There is a huge amount of evidence supporting the perioperative use of local anesthetics, either administered systemically or used as part of regional anesthesia techniques for a variety of reasons: the drugs and their associated analgesic procedures are effective regarding pain relief due to their ability to block the voltage-gated sodium channel, thus inhibiting nerve cell depolarization [1,2], they might reduce postoperative nausea and vomiting (PONV) [3,4], and they might pave the way to an early and enhanced recovery after surgery [5]. Additionally, by using local anesthetics and regional anesthesia, opioid consumption might be reduced, thus leading to improved postoperative bowel function, less constipation, and early restoration of oral nutrition [6]. Moreover, a reduction of postoperative morbidity by dampening the surgical stress response, which could be correlated with perioperative myocardial infarction, pulmonary infection, and thromboembolism, might be another strong advantage of the use of regional anesthesia and LA [7,8].

However, there is more to it. The hypothesis that regional anesthesia and LA might be able to influence cancer recurrence was generated from retrospective studies, and over the last couple of years, several experimental studies—both in vivo and in vitro—have pointed out the importance of the anti-inflammatory and even anti-cancer/anti-metastatic effects of LA in this context and have provided insight into potential mechanisms by which the LA might be able to exert their impact on malignant cells. This narrative review will highlight and summarize the current knowledge regarding these potentially beneficial anti-metastatic effects of the LA in order to increase the acceptance of this concept among our fellow anesthesiologists.

## 2. Background Concept: Circulating Tumor Cells

Since 1869, when Thomas Ashworth described malignant cells in the peripheral blood of a patient with metastatic subcutaneous thoraco-abdominal tumors [9], this particular cell population, shed from the primary tumor into the bloodstream, was termed as circulating tumor cells (CTCs) and has gained a lot of scientific attention over the years and decades. During surgical procedures, CTCs are released into bloodstream [10] and—depending on the tumor entity—prognosis might be negatively correlated with their quantity [11]. Techniques for precise detection and characterization have evolved more in recent years [12]. Some of these techniques are based on the detection of epithelial surface markers such as EpCAM, which is highly expressed, for example, by breast cancer cells [13]. Other methods, such as immunocytochemical characterization, which separate CTCs by their distinct morphological features, are also available [14]. As already mentioned, the detection of CTCs in an individual patient’s blood as well as the number of CTCs has been correlated with metastasis, disease status [15], and clinical outcome, for example, in HER-2-positive breast cancer [16] or thyroid cancer [17]. However, before becoming CTCs, the epithelial tumor cells have to undergo epithelial-to-mesenchymal transition (EMT) to be able to migrate, invade their surroundings, and finally enter the circulation [18]. After exiting the blood stream at a remote location, the cells might then become epithelial again (mesenchymal-to-epithelial transition, MET) and form new metastatic sites [19] (see also Figure 1).

It could even be assumed that CTCs might be able to form new metastatic sites even after complete tumor excision by so-called tumor self-seeding. Kim and colleagues investigated the ability of malignant human breast, colon, and melanoma cells to seed a tumor from circulation in mouse model, showing that self-seeding might only require minimal adaptation of CTCs to the recipient microenvironment [21]. The authors could further demonstrate that CTCs sense attraction signals from the tumor and are, furthermore, able to extravasate and invade the surrounding tissue in response to these signals [21].

Although controversially discussed, several clinical trials were able to show that CTCs might serve as a potential prognostic marker, for example, in colorectal cancer [22,23]. Even the use of CTCs in monitoring tumor response to systemic therapy is widely examined. Hou and colleagues analyzed 97 blood samples from patients suffering from small-cell lung cancer using the earlier mentioned EpCAM-based immunomagnetic detection method [24]. This particular study demonstrated the importance of both baseline CTC numbers as well as of the changes in CTC numbers after chemotherapy as prognostic factors in patients with small-cell lung cancer [24].

It has been demonstrated that CTCs are increasingly released during the crucial perioperative period [25], the short stretch of time in which the anesthesiologist is responsible for the patient. Increasing evidence suggests that this period might be crucial regarding the long-term outcome after cancer surgery, possibly—among other factors—due to a significant inflammatory response, which then impairs the ability of the innate immune system to detect and destroy CTCs [26,27]. The innate immune system, especially the natural killer (NK) cell activity (NKA), is significantly unstable under stress [28]. Under normal circumstances, NK cells are mainly responsible for a phenomenon called immune surveillance, which also includes the detection of CTCs [26]. A significant loss of NKA after abdominal surgery, for example, leads to a compromised resistance to tumor development in rats; Ben-Eliyahu and colleagues were able to show an increased lung tumor retention in rats, which received surgery prior to inoculation with radiolabeled tumor cells in comparison to the unstressed control group [29]. An attenuation of this stress response, e.g., by local anesthetics, might therefore, in turn, be able to decrease the ability of CTCs to metastasize [26,30].

## 3. Background Concept: Inflammatory Stress Response

### 3.1. Overview

Appropriate activation of the innate and adaptive immune response requires a balanced and sufficient cytokine production. Under normal circumstances, i.e., in the absence of stress, the immune system is able to avoid a hyper-inflammatory response [31,32]. The current SARS-CoV-2 pandemic has again impressively demonstrated how a dysregulation of cytokine production with excessive high circulating levels of these biological messengers can cause systemic collateral damage leading to multi-organ-failure [31]. However, not only pathogens or autoimmune disorders are able to trigger the so-called cytokine storm [31]. Cancer and its therapies, such as surgery or chemotherapy, may as well cause an immune cell hyperactivity [33,34,35]. As already described, the perioperative period as a vulnerable time has become a focus of attention in clinical and experimental studies over the past decades—especially in cancer surgery [36]. Surgical trauma inevitably leads to an inflammatory response [37], unbalancing pro- and anti-inflammatory factors [38], ultimately leading to additional immunosuppression [39], which in turn might then favor the CTCs to escape their immune surveillance [26] (Figure 2). Tissue trauma causes the release of vasoactive meditators (e.g., leukotrienes and histamine) and plasma components evoking an inflammatory microenvironment and systemic acute phase reaction with the release of pro-inflammatory cytokines like interleukin (IL)-1, IL-6 [40,41], IL-8, and tumor necrosis factor α (TNF-α) [42]. These cytokines, however, have distinct effects; high levels of IL-6, for example, lead to the initiation of signal transduction processes, ultimately causing endothelial hyperpermeability and hypotension, an effect that might be important during the pathogenesis of acute lung injury and pulmonary edema [43]. TNF-α not only induces fever and augments systemic inflammation, but also regulates parts of the immune system, e.g., by inducing inflammatory signaling events involving nuclear factor kappa B (NF-kB) [44]. The activation of this transcription factor leads to an increased expression of pro-inflammatory genes, thus further enhancing other inflammatory processes [45].

### 3.2. Inflammation, the Inflammatory Response, and Cancer

Inflammation affects disease progression. Virchow hypothesized in 1863 that injury and chronic inflammation might serve as the origin of tissue proliferation and cancer. Although the exact mechanisms are still not fully understood, there is increasing evidence that an inflammatory environment enhances proliferation, survival, migration, and angiogenesis of tumor cells and that misguided immune cell recruitment might be related with cancer recurrence [47]. Kim and colleagues were able to provide evidence for the assumption that IL-6 and IL-8 might be tumor-derived attraction signals [21]. High IL-6 serum-levels are associated with poor prognosis for lung, breast, and colon cancer [48,49,50]. TNF-α-induced NF-kB activation seems to play a major role during these pathophysiologic circumstances as well [51]; upregulated NF-kB transcription might be a critical link between inflammation and cancer, as inactivation of the NF-kB pathway attenuates the formation of inflammation-associated tumors in a colitis-associated cancer model in mice [52]. Another important molecule in terms of cancer and inflammation is intercellular adhesion molecule 1 (ICAM-1). It usually serves as a counter receptor for the neutrophil cluster of differentiation (CD) 11b/CD18 on the surface of endothelial cells, but is also expressed by many cancer cell types [53,54]. A tight adherence of neutrophils and tumor cells via the tumor-expressed ICAM-1 activates the neutrophils, weakens the endothelial barrier, and enhances the extravasation of CTCs [55]. Vascular hyper-permeability and endothelial barrier function is mainly regulated by Src tyrosine protein kinase (Src) [56], which also plays an important role for the metastatic potential of tumor cells, due to its ability to regulate, e.g., tumor cell migration and invasion by various signal transduction pathways, including TNF-α [57].

### 3.3. Anti-Inflammatory Effects of Local Anesthetics

Local anesthetics, especially the amide local anesthetics, have strong anti-inflammatory properties, which have also been studied extensively [42]. Lidocaine and ropivacaine, for example, were demonstrated to be able to preserve endothelial barrier function by an attenuation of TNF-α-induced Src activation in vitro, e.g., in pulmonary endothelial cells [58], which also leads to reduced phosphorylation of ICAM-1 and diminished neutrophil adhesion [58]. Two further studies reported a beneficial effect of ropivacaine on experimental lung injury in rats and mice, which was also due to a reduction in Src activation as well as in ICAM-1 expression [59,60]. A decreased phosphorylation of IkB by lidocaine and, therefore, an inhibition of NF-kB activation as observed by Lang and colleagues in epithelial cells in vitro underlines the anti-inflammatory effects of the LA once more [61]. It could also be shown that lidocaine and bupivacaine, another amide LA, are able to inhibit the release of leukotriene B4, IL-1 [62], and IL-8 [63] in vitro. Lan and colleagues could even demonstrate an attenuation of IL-1*β*, IL-6, and IL-8 by lidocaine in activated human umbilical vein endothelial cells after TNF-α stimulation under ischemia/reperfusion-injury conditions [64]. Similar results could be observed in a recent clinical study evaluating patients undergoing laparoscopic cholecystectomy; patients who had received an intravenous lidocaine infusion had decreased postoperative serum levels of IL-1, IL-6, interferon γ. and TNF-α when compared to an infusion with normal saline [65].

Most of these effects were observed at clinically relevant, non-toxic concentrations of the drugs. However, although these concentrations might also be reached in plasma via absorption of LA after a regional anesthesia procedure, the experimental data suggest that the systemic use of the drugs might be favorable in terms of the anti-inflammatory properties of the drugs [2,58].

## 4. Local Anesthetics and Cancer–What Do We Know So Far?

### 4.1. Historic Clinical Data

Several retrospective studies reported a possible beneficial effect of LA on the outcome, i.e., the overall or recurrence-free survival of patients after tumor surgery. One of the first reports from Exadaktylos and colleagues retrospectively analyzed metastasis-free-survival of 129 women with breast cancer undergoing mastectomy and axillary clearance [66]. Patients treated with paravertebral anesthesia plus general anesthesia had a significant advantage regarding their recurrence-free survival at 12 months compared to women who received general anesthesia only (94% vs. 82%) [66]. This particular study gave rise to several further retrospective analyses. Biki and colleagues focused their analysis on prostate cancer; they compared patients with invasive prostatic carcinoma undergoing surgery between 1994 and 2003 receiving either general anesthesia plus epidural analgesia or general anesthesia with opioid analgesia only and were able to show that the epidural group had an 57% lower risk of recurrence compared with general anesthesia and opioids, even after adjusting the groups towards tumor size, Gleason Score. and prostate-specific antigene [67]. However, after these first encouraging results, there were also several studies reporting no effect; Cummings and colleagues, for example, could not show a difference between patients receiving an epidural analgesia in addition to general anesthesia or not regarding recurrence or survival after resection of gastric cancer [68]. A large randomized trial by Myles and colleagues also concluded that there was no association by the use of epidural anesthesia and cancer-free survival in 503 patients undergoing abdominal surgery [69]. It has to be noted, however, that this particular study was not powered to detect a difference regarding cancer recurrence. Additionally, most studies evaluated cancer recurrence after regional anesthesia. In accordance with the pre-clinical data outlined above, it might be reasonable to hypothesize that the systemic use and application of the LA might be able to exert more pronounced effects.

Several theories regarding possible mechanisms of the observed potential beneficial effects of the LA results have been stated:(1)It is well-known that the use of regional anesthesia and LA might lower the use of opioids or volatile anesthetics during general anesthesia [70]. Several studies also suggested that these drugs and anesthetics might promote cancer progression and reduce long-term survival [71], maybe by promoting tumor angiogenesis [72,73]. However, a more recent experimental study evaluating the effect of opioids in a mouse model of breast cancer surgery reported no negative impact of morphine on the progression of the disease [74]. Negative effects of opioids and volatile anesthetics on the NKA have also been observed and might, therefore, also be important in this regard [30,75]. The reduction in opioids and volatile anesthetics was one of the first possible explanations of the observed beneficial clinical effects of the perioperative use of LA in patients undergoing surgery. However, given the more recent evidence—including studies evaluating the effects of sevoflurane, e.g., in breast cancer [76]—this hypothesis is more likely to be incorrect.(2)As outlined above, there is strong evidence that LA and regional anesthesia might be able to reduce perioperative inflammation and the stress response as induced by surgery [65,77], and also preserve NKA as one of the most important factors for the detection and destruction of CTCs [78,79]. This systemic effect of the LA might, therefore, have a possible positive impact on perioperative processes leading to new micro-metastases, e.g., by CTCs, thus allowing a prolonged (at least recurrence-free) survival.(3)As the theories regarding the indirect effects, induced by a reduction of potentially harmful circumstances as presented above are not able to completely explain the observed effects in cancer patients, several—mostly experimental—studies examined potential direct effects of LA on malignant cells and CTCs as outlined in the next chapter of this article.

### 4.2. First Experimental Data

After the first encouraging results from the retrospective analyses had been published, researchers tried to provide evidence for potential beneficial (direct) effects on malignant cells and CTCs. Most of these observed effects were—at least in part—due to the already mentioned anti-inflammatory effects of LA and suggested that a systemic administration would be much more important than a local effect at the site of injection. Lidocaine and ropivacaine, for example, inhibited TNF-α-induced Src activation independent of sodium channel blockade in non-small cell lung cancer cells in vitro, thus also reducing tumor cell migration [80]. Further downstream of these signaling events, it appears that both drugs are also be able to inhibit TNF-α-induced signaling events involving focal adhesion kinase and caveolin-1, which explained an also observed reduction in the release of matrix-metalloproteinase (MMP)-9 [81]. MMPs are enzymes utilized by the tumor cells to break up the extracellular matrix in order to invade the surrounding tissue [82]. The inhibition of these signal transduction events ultimately leads to a reduction in the TNF-α-induced invasiveness of the tumor cells in this study [81].

Tumor growth and apoptosis might also be affected by LA. Treatment of human breast cancer cell lines MCF-7 and MCF-10A with lidocaine and bupivacaine in clinically relevant concentrations revealed an inhibition of cell viability and an induction of apoptosis-related proteins in vitro [83]. Potentially beneficial actions were examined in various different tumors. Xuan and colleagues investigated that bupivacaine possesses an anti-metastatic and anti-proliferative effect on human ovarian and prostate cancer cell lines [84]. There is also experimental data underlining that LA can inhibit the growth of human hepatocellular carcinoma cells [85]. Both lidocaine and ropivacaine can affect the expression of cell-cycle-related genes and induce apoptosis in these cells [85]. It has also been demonstrated that LA might slow down cancer cell growth in vitro and induce cell death at the same time in pancreatic [86] and colon cancer cells [87]. The drugs might be able to directly induce apoptosis via the mitochondrial and p38 mitogen-activate protein kinase MAP-kinase-dependent pathways as Lu and colleagues found out after incubating a neuroblastoma cell line with bupivacaine [88].

## 5. Recent and Current Data

### 5.1. Experimental Studies

Several research groups have put a lot of effort into a further exploration of these initially presented mechanisms by which the LA might exert their beneficial effects.

Most interestingly, newer evidence points towards a possible synergistic effect of the LA together with chemotherapy; in vitro, lidocaine appears to have an enhancing effect on the chemotoxicity of cisplatin via the demethylation of retinoic acid receptor beta 2 (RARβ2) located in the cell nucleus and tumor suppressor Ras association domain-containing protein 1 (RASSF1) in breast cancer cells [89]. Following these in vitro results, it has subsequently also been shown that lidocaine alone can reduce the tumor size of hepatocellular carcinoma and, moreover, it enhances the sensitivity of the tumor cells against cisplatin in an in vivo murine model [90]. A study from Freeman and colleagues also found a potential metastasis-inhibiting effect of perioperative systemic lidocaine combined with cisplatin in another murine model of triple negative breast cancer with a Stage IV metastatic burden [91]. In this study, 50 animals were treated with cisplatin only or with cisplatin plus lidocaine at clinically relevant concentrations (bolus of 1.5 mg/kg plus 2 mg/kg x h^−1^ during surgery) and lidocaine treatment lead to fewer metastatic lesions in the animals’ lungs [91]. The same scientific research group published a study in 2019 focusing on a four-branched murine 4T1 model on the influence of lidocaine, methylprednisolone, and propofol in combination with general anesthesia with sevoflurane on pulmonary metastasis after 14 days after surgery of the primary breast cancer tumor [92]. Here, the hepatic metastasis load was equal in all groups. However, lidocaine and propofol each reduced the post-mortem in vitro cultured pulmonary metastasis colonies [92]. These findings are consistent with another study comparing the influence of lidocaine in dependency of sevoflurane compared to ketamine/xylazine anesthesia on pulmonary metastasis in a 4T1 mouse model of breast cancer [93]. Of note, the administration of high dose steroids in the 2019 study by Freeman and colleagues [92] even enhanced the pulmonary metastasis burden. The authors hypothesized that these results might be due to a facilitation of the dispersion and metastasis of CTCs by the drug [94].

In another recent study by Chamaraux-Tran and colleagues, the effect of a lidocaine treatment was tested in breast cancer cells in vitro and in an in vivo mouse model [95]. In accordance with earlier results, there was a direct cytotoxic effect on the tumor cells. The triple-negative cell lines especially were more sensitive to the treatment with lidocaine. Lidocaine had an inhibitory effect on breast cancer cell migration with a predominant effect (again) on the triple-negative cell line. In the mouse experiments, the lidocaine group showed a delay in the development of peritoneal carcinomatosis after injection of MDA-MB-231 cells and repeated injections of the drug into the peritoneal cavity. Of note, no intravenous administration has been examined, but instead, the authors claim a potential beneficial peritoneal administration (in a dose already used for shoulder pain after laparoscopy) to also use the direct cytotoxic effects of lidocaine in case of peritoneal dissemination of tumor cells [95].

Following the data showing that LA are able to induce apoptosis in (for example breast and thyroid) cancer cells by activating caspases and regulating the mitogen-activated protein kinase signaling pathway [96], the apoptosis-inducing effect has also been shown in hepatocellular cancer cells [90]. In a recent study also in lung cancer cells, a potentially beneficial effect of lidocaine treatment on cell viability and proliferation has been shown [97]. Here, the authors hypothesized that the mechanism of the lidocaine-anti-tumor-effect might be based on an up-regulation of miR-539 microRNA, which inhibits signaling of the epidermal growth factor receptor (EGFR) through direct binding, thus regulating downstream signaling via extracellular-signal regulated kinase (ERK) and the phosphatidyl-inositol 3-kinases (PI3K)/AKT pathway [97]. The activation of the latter has been found in melanoma cells [98,99] and is known to be blocked by amide LA [58,80]. Another recent study evaluating cell viability of breast cancer cells also found an inhibitory effect of different LA [100]. However, most of the effects in this particular study were only observed at concentrations ten times higher than the clinically relevant plasma concentration. The observed cell alterations might, therefore, be considered to be due to toxic rather than pro-apoptotic effects of the LA in this setting.

The inhibiting effect of lidocaine on Src tyrosine protein kinase (Src) [101,102,103] indicates that systemically administered local anesthetics might potentially be able to inhibit cancer cell metastasis [26,96,104,105]. A recent study by Wall and colleagues in 2019 supported this hypothesis by measuring the metastatic burden in lung and liver as well as MMP-2 levels in dependency of the treatment with lidocaine and the Src-inhibitor bosutinib in a 4T1 mouse tumor model [106]. Bosutinib neutralized the protecting effect of lidocaine regarding lung metastasis and levels of MMP-2. However, the authors claimed that it remains unclear if the findings are due to a direct Src effect or another pathway [106].

It has also been shown that lidocaine at clinically relevant doses might have demethylating effects on breast cancer cell lines [107,108], while at the same time enhancing the effect of the deoxycytidine analog chemotherapeutic decitabine [108]. Besides the therapeutic effects in acute myeloid leukemia, decitabine is a possible agent in the therapy of breast cancer as well and could be used as a second line therapy in chemotherapy-resistant patients [109,110]. Thus—at least based on the results of these experimental studies—the additional administration of intravenous lidocaine might be able to enhance its therapeutic effect and might be of interest for future studies.

Most of the outlined anti-inflammatory and potentially anti-metastatic effects of the LA are mediated independent of sodium channel blockade [80]. However, it has also been observed that lidocaine (in clinically relevant doses) is able to block cancer-associated and prognostic relevant variants of voltage-activated sodium channels like Nav1.5 [111]. Interestingly, a recent animal study detected chronic electric activity in solid breast tumor masses in mice [112]. This activity is supposedly of neuronal origin, as it has a connection to the parasympathetic nervous system, which disrupts on injection of lidocaine and chemical sympathectomy [112]. With these neuronal networks, systemically administered lidocaine could, therefore, have a new target in its first described mechanism of action, thus disrupting the neuronal membrane potential and neural activity during tumor growth and metastasis.

### 5.2. Clinical Studies–Systemic Use of Local Anesthetics

Clinical studies examining the short- and long-term effects of (amide) LA on perioperative pain und its chronification continued to increase in number over the past few years. The authors of a recently updated Cochrane analysis, however, were uncertain whether intravenous lidocaine might have beneficial effects on postoperative pain, nausea, or opioid consumption [113].

As the use of intravenous lidocaine is still considered an off-label use in most countries, concerns regarding the risk of intoxications after systemic administration of the drug have been raised repeatedly and were also addressed in a very recent consensus paper [114]; here, the authors provided evidence for the fact that intravenous lidocaine might be considered safe, if clinicians followed several precautions, including correct dosage and a 24 h limit for the duration of the drug infusion, as well as close post-operative monitoring [114], the latter possibly bearing the potential to collide with enhanced recovery after surgery (ERAS) programs, in which intravenous lidocaine has already been established as part of the multimodal analgesic regimen [115,116].

Due to these concerns raised by clinicians, the perioperative systemic use of LA should always be a “risk-benefit” decision depending on the individual patient, her/his co-morbidities, the surgical procedure, and, of course, the available evidence [114].

### 5.3. Clinical Studies–Local Anesthetics and Cancer Recurrence

A very recent RCT compared the rate of breast cancer recurrence after curative surgery in more than 2000 patients receiving either a propofol-based anesthesia in combination with a paravertebral nerve block or a general anesthesia with sevoflurane and an opioid-based analgesic regimen. Unfortunately, there was no difference regarding the primary outcome between these two groups [117]. However, this particular and well-executed study once more underlines the importance for more clinical studies evaluating the potential impact of the systemic use of the drugs in terms of their anti-inflammatory or even anti-metastatic effects. These prospective clinical trials focusing on the outcome of patients undergoing cancer surgery with or without systemic administration of LA are still lacking. Not only, since the 2014 Cochrane review [118], are we aware of the conflicting—and not very convincing—data regarding the impact of the perioperative use of regional anesthesia in cancer patients, but some studies have found an effect in some types of cancer [66,67,119,120,121,122], and some studies have not [69,123,124,125,126] or only in certain cancers in subpopulations [127]. In various articles, the urgent need for clinical trials evaluating the effects of perioperative, systemic administration of LA, e.g., of lidocaine during the perioperative period of cancer patients, is stressed [26,128,129].

Following this call, there are several clinical trials currently investigating a potentially beneficial effect of lidocaine in cancer patients. However, unfortunately, some of these studies do not focus on the anti-metastatic effect of systemically administered lidocaine and follow a more clinical approach and outcome protocol (NCT00938171, NCT03824808, NCT03530033) or do not compare LA vs. placebo (NCT03134430). Some of the studies are still promising though; a currently recruiting double blinded randomized placebo-controlled clinical trial (NCT04048278) is designed to compare the effects of lidocaine infusions on Src activity in CTCs during the perioperative period in patients undergoing robotic surgery for pancreatic cancer. Another clinical trial (NCT04162535) is recruiting 40 patients and focuses on the secondary end points on the survival comparison of intravenous lidocaine in combination with a propofol-based total intravenous anesthesia (TIVA) compared to standard treatment (TIVA without lidocaine or sevoflurane-based general anesthesia) in colorectal cancer surgery. Planned in much larger (n = 450) dimensions is a quadruple-blinded and randomized clinical trial (NCT02786329) in colorectal cancer, again comparing intravenous lidocaine in combination with TIVA, sevoflurane versus TIVA, or sevoflurane alone. This particular trial mainly aims to investigate survival after surgery and the incidence of recurrence within the first 5 years following surgery. A clinical trial (NCT02839668) in breast cancer patients has already completed the recruitment phase and investigates the use of intravenous lidocaine (1.5 mg/kg) in addition to either TIVA or sevoflurane. This study focuses on the levels of vascular endothelial growth factor A (VEGF-A) and postoperative pain, as well as patients’ survival and VEGF-receptor density.

In addition, several clinical trials are planned and registered but currently not yet recruiting, including the VAPOR-C trial (NCT04316013); here, the investigators are planning to include a total of 5736 participants with colorectal or non-small cell lung cancer. The four different treatment arms of the study will hopefully be able to assess the effect of the choice of anesthetic (sevoflurane vs. propofol) and the impact of perioperative lidocaine infusions (versus placebo).

## 6. Conclusions

Although a large amount of experimental evidence points towards a potential beneficial effect of the perioperative use of regional anesthesia and local anesthetics—preferably administered systemically—the exact role and impact of the use of these substances in the setting of cancer surgery is still unclear, mostly due to the lack of clinical data coming from randomized controlled trials. As several clinical trials evaluating the effect of local anesthetics in patients undergoing cancer surgery are currently recruiting patients, we are eagerly awaiting these results in order to answer this important research question in the field of anesthesia.

## Figures and Tables

**Figure 1 jcm-10-00719-f001:**
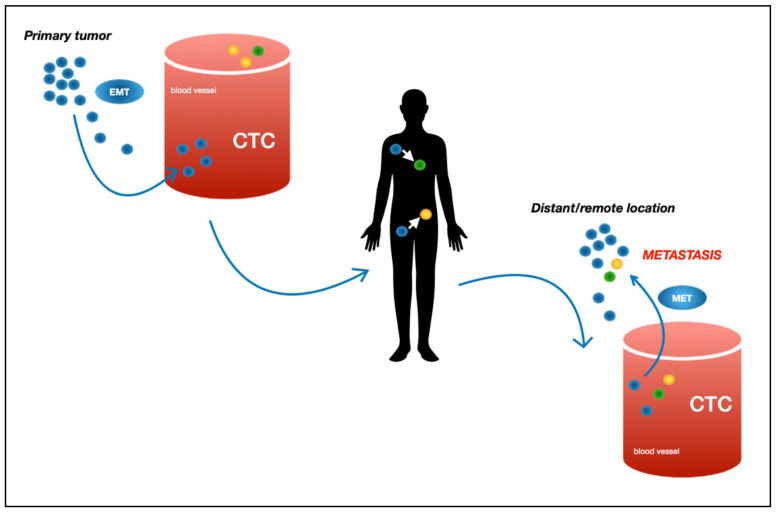
Schematic illustration of metastasis formation by circulating tumor cell. CTC = circulating tumor cells; EMT = epithelial-to-mesenchymal transition; MET = mesenchymal-to-epithelial transition [20].

**Figure 2 jcm-10-00719-f002:**
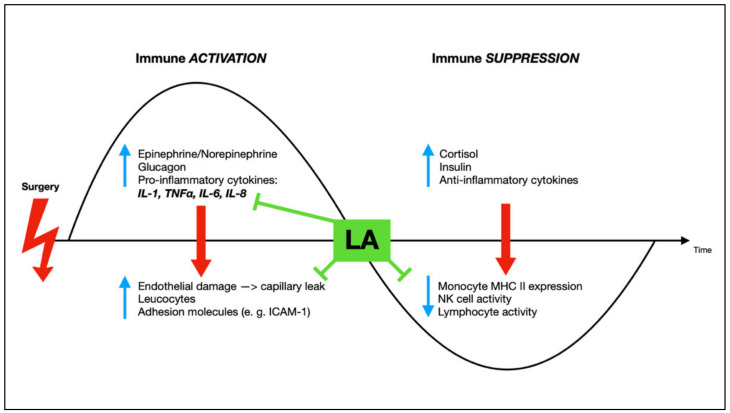
Schematic illustration of processes leading to perioperative inflammation and immune suppression (surgical stress response) and possible inhibition by local anesthetics (shown as ⟂). LA = local anesthetics; IL = interleukin; TNFα = tumor necrosis factor α; ICAM-1 = intercellular adhesion molecule 1; MHC = major histocompatability complex; NK cell = natural killer cell. Modified after [46].

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
