# Peer review of "Local Anesthetics and Recurrence after Cancer Surgery-What’s New? A Narrative Review"

_jcm, 2021, doi:10.3390/jcm10040719_

Round 1

Reviewer 1 Report

  1. Typos in the manuscript such as p3, line 116; p4, line 154
  2. Please comment on the most recent published perspective study of regional anesthesia on breast cancer recurrence in lancet 2019. Pubmed Id:31645288
  3. Please comment on a recent study of local anesthetics on breast cancer pubmed ID:29914426
  4. The content under subtitle "4.3 Recent and current data" are not well organized and readers are easy to lost through reading. Please rearrange all the evidence based on study types or the aspects of impact local anesthetics induced. More importantly, please evaluate those evidence and provide insight into this field. A illustrative diagram or table might help 

Reviewer 2 Report

This is obviously quite a difficult area to write a review at the moment, due to the lack of specific evidence in a number of areas, and the rapidly changing evidence in others.  The main difficulty that I have with the paper, is that I think that the evidence with regards to regional anaesthesia/analgesia and the systemic effects of local anaesthetics has been muddled together a bit.  Obviously there is some cross over given the likely systemic absorption of local anaesthetic with regional anaesthetic techniques, but I think that the likelihood that regional anaesthesia reduces tumour recurrence has significantly reduced since the recent publication of the Sessler et al trial( Lancet 2019 Nov 16;394(10211):1807-1815. Recurrence of breast cancer after regional or general anaesthesia).  The evidence presented with regard to systemic effect of local anaesthetic is reasonable, but the references to different types of cytokines does make it a bit difficult to follow.  May be a graphical representation of the immune system response with arrows to indicate the potential interaction of local anaesthetics would help improve readibility.  The english is reasonable, but I would have someone proof read it before resubmission, as things like starting the paragraph with "But..." second paragraph first page, don't sit that well with native speakers.  So overall, reasonable effort, but try and separate out the systemic and regional anaesthesia effects further, and add in graphics to help explain the immune response and effect of local anaesthetics.

Reviewer 3 Report

Thank you for giving me the opportunity to review this paper. The field of onco-anesthesiology if rapidly evolving, and this is one of the major topics within the field of anesthesia.

This narrative review covers the effects of local anesthetics on the immune system and metastasis formation. It touches upon both regional anesthesia and intravenous local anesthetics. While there are other reviews published about this topic in recent years, this article summarizes the current status of research within this field of onco-anesthesia. It can appropriately supplement articles submitted to the special issue in Journal of Clinical Medicine. The paper is well structured and readable with thorough descriptions of the pathophysiologic effects of local anesthesia and cancer. I have very little to add.

The current evidence is nicely summarized. It is mainly based on observational and laboratory studies. Yet, since ongoing RCT’s are mentioned, I think it would be worth mentioning the large trial by Sessler et al (Lancet 2019) randomizing 2000 patients undergoing breast cancer surgery to either sevoflurane + opioids or TIVA + paravertebral nerve block. They did not find an effect on cancer recurrence; possibly because the surgical stress of these operations is minor compared with other types of surgery. Therefore, a discussion of these results would be of benefit for the reader.

As this is submitted to a clinical journal, I would recommend decribing implementation of intravenous anesthesia in a clinical setting. There are challenges in administration of intravenous lidocaine after surgery, that could be discussed e.g. need of continous cardiac monitoring which to some degree compromizes enhanced recovery after surgery (ERAS).

Further, there is an ongoing discussion in anesthesiology about concerns of toxicity for intravenous lidocaine. The therapeutic index is narrow with high risk of adverse events (of which one death was reported in UK), and moreover, the clinical analgesic effect seem to be quite limited (Foo et al, anesthesia 2021, Pandit et at, Anaesthesia 2021). Therefore, I find it appropriate that this paper focuses on the antimetastatic effects of local anesthesia instead of postoperative analgesia and recovery. However, since it is stated in the introduction that one purpose is to "increase acceptance of this concept among our fellow anesthesiologists", I think that these concerns should be discussed in a paper published in a clinical journal.

Lastly, the title is "local anesthesia and survival after cancer surgery…”. Since the describes the effect of local anesthetics on the process of cancer progression, would it be more appropriate to use the word “recurrence” instead of “survival”?

Round 2

Reviewer 2 Report

I think that the changes made have improved the paper, no further revision required.